# Automatic discovery of cell types and microcircuitry from neural connectomics

**Eric Jonas[1]\*, Konrad Kording[2,3,4]**

[1]Department of Electrical Engineering and Computer Science, University of California, Berkeley, Berkeley, United States; [2]Department of Physical Medicine and Rehabilitation, Northwestern University, Chicago, United States; [3]Department of Physical Medicine and Rehabilitation, Rehabilitation Institute of Chicago, Chicago, United States; [4]Department of Physiology, Northwestern University, Chicago, United States

**Abstract** Neural connectomics has begun producing massive amounts of data, necessitating new analysis methods to discover the biological and computational structure. It has long been assumed that discovering neuron types and their relation to microcircuitry is crucial to understanding neural function. Here we developed a non-parametric Bayesian technique that identifies neuron types and microcircuitry patterns in connectomics data. It combines the information traditionally used by biologists in a principled and probabilistically coherent manner, including connectivity, cell body location, and the spatial distribution of synapses. We show that the approach recovers known neuron types in the retina and enables predictions of connectivity, better than simpler algorithms. It also can reveal interesting structure in the nervous system of *Caenorhabditis elegans* and an old man-made microprocessor. Our approach extracts structural meaning from connectomics, enabling new approaches of automatically deriving anatomical insights from these emerging datasets.

\*For correspondence: jonas@ eecs.berkeley.edu

**Competing interests:** The authors declare that no competing interests exist.

## Introduction

Emerging connectomics techniques (*Zador et al., 2012*; *Morgan and Lichtman, 2013*) promise to quantify the location and connectivity of each neuron within a tissue volume. These massive datasets will far exceed the capacity of neuroanatomists to manually trace small circuits, thus necessitating computational, quantitative, and automatic methods for understanding neural circuit structure. The impact of this kind of high-throughput transition has been seen before—the rise of sequencing techniques necessitated the development of novel computational methods to understand genomic structure, ushering in bioinformatics as an independent discipline (*Koboldt et al., 2013*).

The brain consists of multiple kinds of neurons, each of which is hypothesized to have a specific role in overall computation. Neuron types differ in many ways, for example, chemical or morphological, but they also differ in the way they connect to one another (*Seung and Sümbül, 2014*). In fact, the idea of well defined, type-dependent local connectivity patterns (microcircuits) has a long history (*Passingham, 2002*), and is prominent in many areas, from sensory (e.g., retina; *Masland, 2001*) to processing (e.g., neocortex; *Mountcastle, 1997*) to movement (e.g., spinal cord; *Grillner et al., 2005*). These types of repeated computing patterns are a common feature of computing systems, even occurring in man-made computing circuits. It remains an important challenge to develop algorithms to use connectivity-based anatomical data (connectomics) to automatically back out underlying microcircuitry.

The discovery of structure is a crucial aspect of network science. Early approaches focused on global graph properties, such as the types of scaling present in the network (*Watts and Strogatz, 1998*).

**eLife digest** The human brain is made up of billions of neurons, which are organised into networks via trillions of connections. The study of the nature of these connections will be central to understanding how the brain works. In recent years, a number of new methods for imaging the brain have made it possible to visualise and map these connections, generating striking images and creating an additional field of neuroscience known as 'connectomics'.

However, the sheer volume of data generated by connectomics is now beginning to exceed the capacity of researchers to analyse it. Just as the advent of genome sequencing required the development of statistical techniques to analyse the resulting data, so the emergence of connectomics has created a need for similarly powerful mathematical models in neuroscience.

Jonas and Kording have developed one such algorithm that can classify the component units of circuits, both biological and man-made, and identify the connections between them. When applied to connectomics data for 950 neurons in the mouse retina, the algorithm generated predictions regarding cell types and patterns of connectivity. The predicted cell types agreed closely with those identified by human neuroanatomists. Results were similarly convincing when the algorithm was applied to the nervous system of the nematode worm and genetic model organism, *Caenorhabditis elegans*, and even when it was asked to classify electronic components and connectivity patterns in a man-made microprocessor.

Algorithms such as that developed by Jonas and Kording will soon be essential for making sense of the vast quantities of data generated by connectomic studies of the human brain. At present, an analysis of 950 neurons requires several hours, thus refinements that make the process faster will likely be required prior to the analysis of larger human datasets. Such algorithms will open up a range of possibilities for examining the structure of the healthy brain, as well as the changes triggered by developmental abnormalities and disease.

While this approach leads to an understanding of the global network, more recent work aims at identifying very small-scale repeat patterns, or motifs, in networks (*Milo et al., 2002*). These motifs are defined not between different node types, but rather represent repeated patterns of topology.

The discovery of structure in probabilistic graphs is a well-known problem in machine learning. Commonly used algorithms include community-based detection methods (*Girvan and Newman, 2002*) and stochastic block models (*Nowicki and Snijders, 2001*). While these approaches can incorporate the probabilistic nature of neural connections (*Hill et al., 2012*), they do not incorporate the additional richer structure present in connectomics data—the location of cell bodies, the spatial distribution of synapses, and the distances between neurons. It is of particular importance that the probability of connections has a strong spatial component, a factor that is hard to reconcile with many other methods. A model attempting to fully capture the variation in the nervous system should take into account the broad set of available features.

When it comes to neuroscience and other computing systems, we expect patterns of connectivity much more complex than traditional motifs, exhibiting a strong spatial dependence arising from the complex genetic, chemical, and activity-based neural development processes.

To address these challenges, here we describe a Bayesian non-parametric model that can discover circuit structure automatically from connectomics data: the cell types, their spatial patterns of interconnection, and the locations of somata and synapses. We show that by incorporating this additional information, our model both accurately predicts the connection as well as agrees with human neuroanatomists as to the identification of cell types. We take as inspiration previous work on identifying cell types automatically from morphology (*Guerra et al., 2011*) and electrophysiology (*Druckmann et al., 2013*).

We primarily focus on the recently released mouse retina connectome (*Helmstaedter et al., 2013*), but additionally examine the *Caenorhabditis elegans* connectome (*White et al., 1986*). Comparing the cell types discovered by the algorithms with those obtained manually by human anatomists reveals a high degree of agreement. We thus present a scalable probabilistic approach to infer microcircuitry from connectomics data available today and in the future.

## Model

We build a structured probabilistic model which begins with the generic notion of a cell being a member of a single type—and these types affect soma depth, distribution of synapses, as well as a cell type and distance-dependent connection probability. For example, retinal ganglion cells may synapse on nearby, but not far away, amacrine cells, with bipolar cells clearly tessellating space and synapsing on both. In machine learning parlance, our method is unsupervised—it seeks to discover structure in data and make predictions in the absence of training data. Rather than taking in examples of types annotated by human neuroanatomists, we instead start with the weakest possible assumption in an attempt to algorithmically discover this structure. We contrast this with the supervised approaches taken in *Guerra et al. (2011)*, where there is high confidence in the (morphologically defined) types and then a supervised classifier is built, as our goal here is explicit discovery of types.

From these assumptions (priors) we develop a generative Bayesian model that estimates the underlying cell types and how they connect. We take as input (*Figure 1A*) the connectivity matrix of cells (*Figure 1B*), a matrix of the distance between cells (*Figure 1C*), the per-cell soma depth (*Figure 1D*), and the depth profile of the cell's synapses (*Figure 1E*). We perform joint probabilistic inference to automatically learn the number of cell types, which cells belong to which type, their type-specific connectivity, and how connections between types vary with distance. We also simultaneously learn the soma depth associated with each type and the typical synaptic density profile (*Figure 1F–H*).

We start with a model for connectivity, the iSBM (*Kemp et al., 2006*; *Xu et al., 2006*), which has been shown to meaningfully cluster connection graphs while learning the number of hidden groups, or types. We extend this approach by adding distance dependence to model salient aspects of microcircuitry via logistic and exponential distance-link functions. We form a unimodal model of cell body depth and a multimodal synapse density profile model (see 'Materials and methods' for mathematical details).

As an illustrative example, consider a network with only three cell types, labeled A, B, and C. Assume these cells are uniformly distributed in space, and that the probability of connection between any two cells, $c_i$ and $c_j$, depends only on their type and their distance, according to a logistic (sigmoidal) function. Let A connect only to nearby B and C cells, but B connect to any C regardless of distance. This is the prior intuition our model is designed to capture.

For the basic link-distance model, we take as input a connectivity matrix $R$ defining the connections between cell $e_i$ and $e_j$, as well as a distance function $d(e_i, e_j)$ representing a (physical) distance between adjacent cells. See the supplemental material for extension to multiple connectivity matrices. We assume there exist an unknown number $K$ of latent (unobserved) cell types, $k \in \{1, 2, 3, \ldots, K\}$, and that each cell $e_i$ belongs to a single cell type. We indicate a cell $e_i$ is of type $k$ using the assignment vector (c), so $c_i = k$. The observed connectivity between two cells $R(e_i, e_j)$ then depends only on their latent type and their distance through a link function $f(\cdot, d(e_i, e_j))$. We assume $f$ is parameterized based on the latent type, $c_i = m$ and $c_j = n$, via a parameter $\eta_{mn}$, as well as a set of global hyperparameters $\theta$, such that the link function is $f(d(e_i, e_j)|\eta_{mn}, \theta)$.

We then jointly infer the posterior distribution of the class assignment vector $(c) = \{c_i\}$, the parameter matrix $\eta_{mn}$, and the global model hyperparameters $\theta$:

$$p(\mathbf{c}, \eta, \theta | R) \propto \prod_{i,j} p\Big( R(e_i, e_j) | f\Big( d(e_i, e_j) | \eta_{c_i c_j} \Big), \theta \Big) \prod_{m,n} p(\eta_{mn}|\theta) p(\theta) p(\mathbf{c}|\alpha) p(\alpha) p(\theta). \tag{1}$$

Our subsequent analysis uses both the full posterior distribution over these parameters as well as the most probable, or maximum a posteriori (MAP), estimate.

For the retina data, we then extend the model with the additional features indicated. Cell soma depth is modeled as a cell-type-dependent Gaussian distribution with latent (unknown) per-type mean and variance. Similarly, each cell has some number $N_i$ of synapses, each of which is drawn from a cell-type-specific density profile with up to three modes.

Inference is performed via MCMC via three composable transition kernels—one for structural, one for per-type parameters, and one for global parameters and hyperparameters. Details of data preprocessing, inference parameters, and runtime can be found in the 'Methods section.

## Metrics

To evaluate the quality of the model fit, we need to use information that quantifies aspects of the data for which we have ground truth information. We focus on two aspects of performance. First, if the

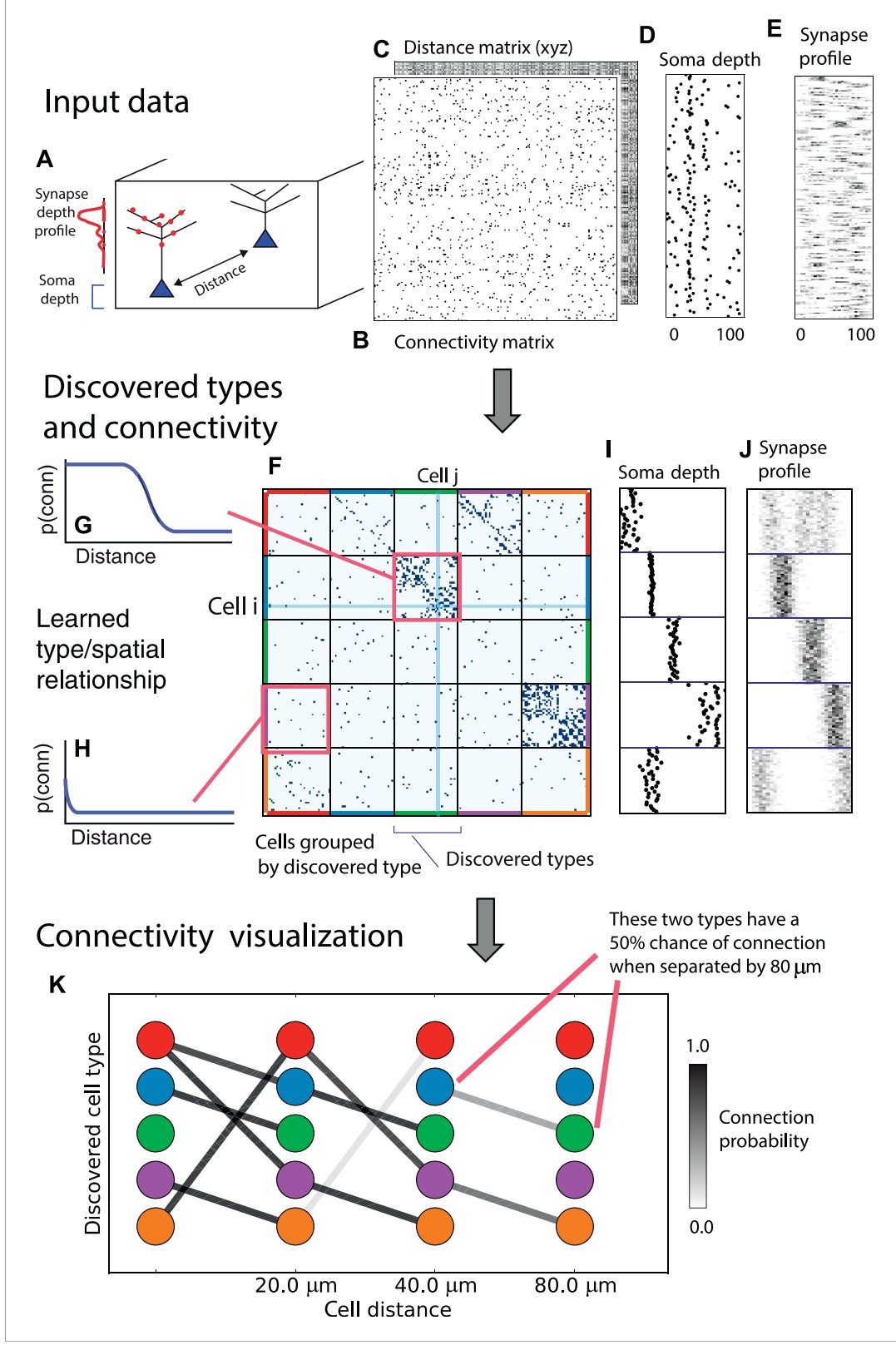

**Figure 1**. Deriving circuitry and cell types from connectomics data. (**A**) As input we take the connectivity between cells (**B**), the distance between them (**C**), the depth of the cell bodies (**D**), and the depth profile of the synapses (**E**). (**F**) Our algorithm discovers hidden cell types in this connectivity data by assuming all cells of a type share a distance-

*Figure 1. continued on next page*

*Figure 1. Continued*

dependent connectivity profile, similar depth, and a similar synaptic density profile, with cells of other types. This results in a clustering of the cells by those hidden types. (**F**) Shows the cell connectivity matrix with cells of the same type grouped together. (**G**) Shows the learned probability of connection (p(conn)) between our different types at various distances—in this case, the cells are likely to connect when they are close. (**H**) Shows the probability of connection (p(conn)) between two cell types that very rarely connect—there is a background 'base' connection rate to account for errors in data, but the probability is very low. (**I**) Shows that we also recover the expected laminarity of types and the depth-specific (**J**) synaptic connectivity. (**K**) We then plot how the connectivity between these types changes as a function of distance between the cell bodies to better understand short-range and long-range connectivity patterns.

model works well, then the probability that a pair of neurons is of the same type should be high if the neurons actually are of the same type. Second, the model should assign a high probability of connection between two cells if they have a connection in the underlying data. We term these two factors clustering accuracy and link-prediction accuracy.

To assess the accuracy of a clustering compared to that determined by neuroanatomists, we employ three metrics—clustering homogeneity, clustering completeness, and the ARI. All metrics equal 1.0 when two clusterings completely agree. Homogeneity reflects the degree to which a found cluster or type contains only a single true type. Completeness measures how much of a true type is contained within a single identified type—a completeness of 1.0 means no true type is split into multiple subtypes. ARI is a metric that reflects both measures (see the supplemental material for more information).

To assess the accuracy of the model for connections, we use link prediction accuracy. If our model accurately captures the true structure of the data, it should be good at predicting if a link exists. We thus train the model on the data with a subset of the links marked as unobserved and thus compute our predictive accuracy. We perform 10-way cross-validation on a given dataset (*Guerra et al., 2011*), learn the resulting model, and use that model to predict the missing synapses. Each potential link between cells is assigned a probability, and we compute the AUC for the resulting ROC curve. An AUC of 1.0 means that we perfectly predict the presence and absence of the missing synapses. We use link prediction accuracy to quantify how good the model is at discovering the underlying connectivity.

## Results

We will first establish that our algorithm works properly and try to understand its properties using simulated data. Subsequently, we will analyze in detail a dataset on the retina. Lastly, we will briefly discuss the analysis of data from the worm *C. elegans* and from an old man-made microprocessor.

### Validation with simulated data where ground truth is known

To validate our model, we performed a series of simulations to test if the model can accurately recover the true underlying network structure and cell type identity. We thus simulate data for which we know the correct structure and compare the estimated structure based on the algorithm (see 'Materials and methods') with the one we used for simulation. We find that the model does a good job of recovering the correct number of cell types (*Figure 2A*), the cell identities (*Figure 2B*), and the spatial extent of each type (*Figure 2C*). For comparison, we show the results using the infinite stochastic block model (iSBM) instead (*Figure 2A–C*, black line) which assumes that only cell type matters, and thus finds small neighborhoods of connected nodes (instead of global connectivity patterns). This contrast shows that while the regular block model can not correctly deal with distance-dependent connectivity, our model can. Our model converges relatively quickly (see 'Mixing of Markov chains') to an estimate of the most probable values for the cell types, which is enabled by using a combination of simulated annealing and parallelized Markov-chain Monte Carlo (MCMC) (see 'Materials and methods' for details). Thus our model at least is promising for application to biological datasets.

### Model mismatch

We next analyze how our model performs in cases where the data are generated with assumptions different from ours. To understand the properties of our model, we attempt connectivity inference on four sets of synthetic data. This helps us understand what our model would do if the data do not obey our assumptions.

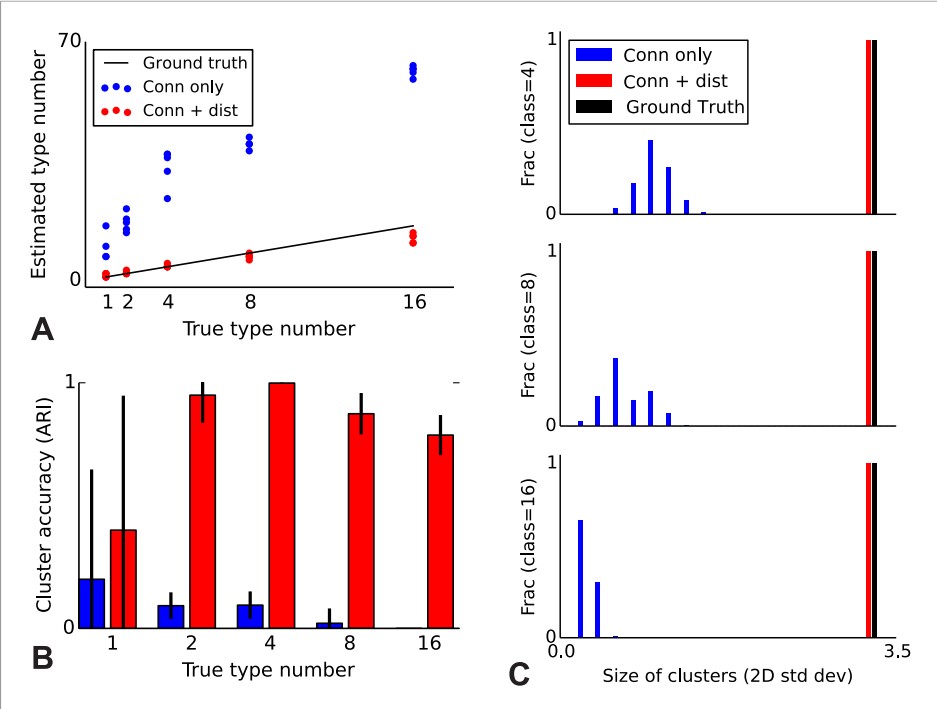

**Figure 2**. Correct recovery of true numbers of hidden types in synthetic data when incorporating spatial information. (**A**) The infinite stochastic block model (which only uses connectivity information) over-estimates the number of classes as it fails to take distance into account, whereas our modeling of the combination of distance and connectivity finds close to the true number of classes. Conn: connectivity; dist: distance. (**B**) As we increase the true number of types, our method continues to find the correct clustering (as measured by the adjusted Rand index, ARI) whereas the infinite stochastic block model (iSBM) overclusters and thus poorly matches ground truth. (**C**) We examine the spatial extent (size) of the discovered types (clusters) by measuring the two-dimensional standard deviation of the cell locations. The y-axis indicates what fraction of the discovered types had a given spatial extent. Without incorporating distance, we identify a large number of small, spatially-localized types. With distance, we see a correct recovery of the spatial extent of each type.

We thus generate 10 sets of synthetic data from each of four existing models. The distance-dependent stochastic block model assumes type depends on distance, the traditional stochastic block model has no notion of distance, the mixed membership block model assumes type is combinatorial, and the latent position cluster model assumes that type is clustered-but-continuous.

If the data are sampled from our model, inference according to our model, unsurprisingly, is good by all measures. It correctly estimates the number of cell types, it is good at predicting connectivity (high area under the curve, AUC), it agrees with human classification (Rand index), it discovers all types, and leads to homogeneous estimates (*Figure 3*, first row). If the data come from a block model without distance dependence, we see that it still does well on all meaningful measures (*Figure 3*, second row). This is unsurprising, as our model learns the distance dependence, even its absence. For the mixed membership model (*Figure 3*, third row), the model grossly overestimates the number of types, by basically allocating a type for each combination of memberships. Otherwise, it still performs relatively well. Lastly, for the latent position clustering model (*Figure 3*, fourth row), the model does poorly. If type is continuous instead of discrete, then our model is basically trying to cover a continuous set with a discrete scenario leading to rather poor performance. However, as we do expect cell types to have a discrete biological basis, we might expect our model to do well with real data.

## Sensitivity to edge effects

Connectomic efforts so far have reconstructed only small sections of neural tissue. Consequently, many connections to cells outside that tissue volume will be lost. We are concerned that this selective

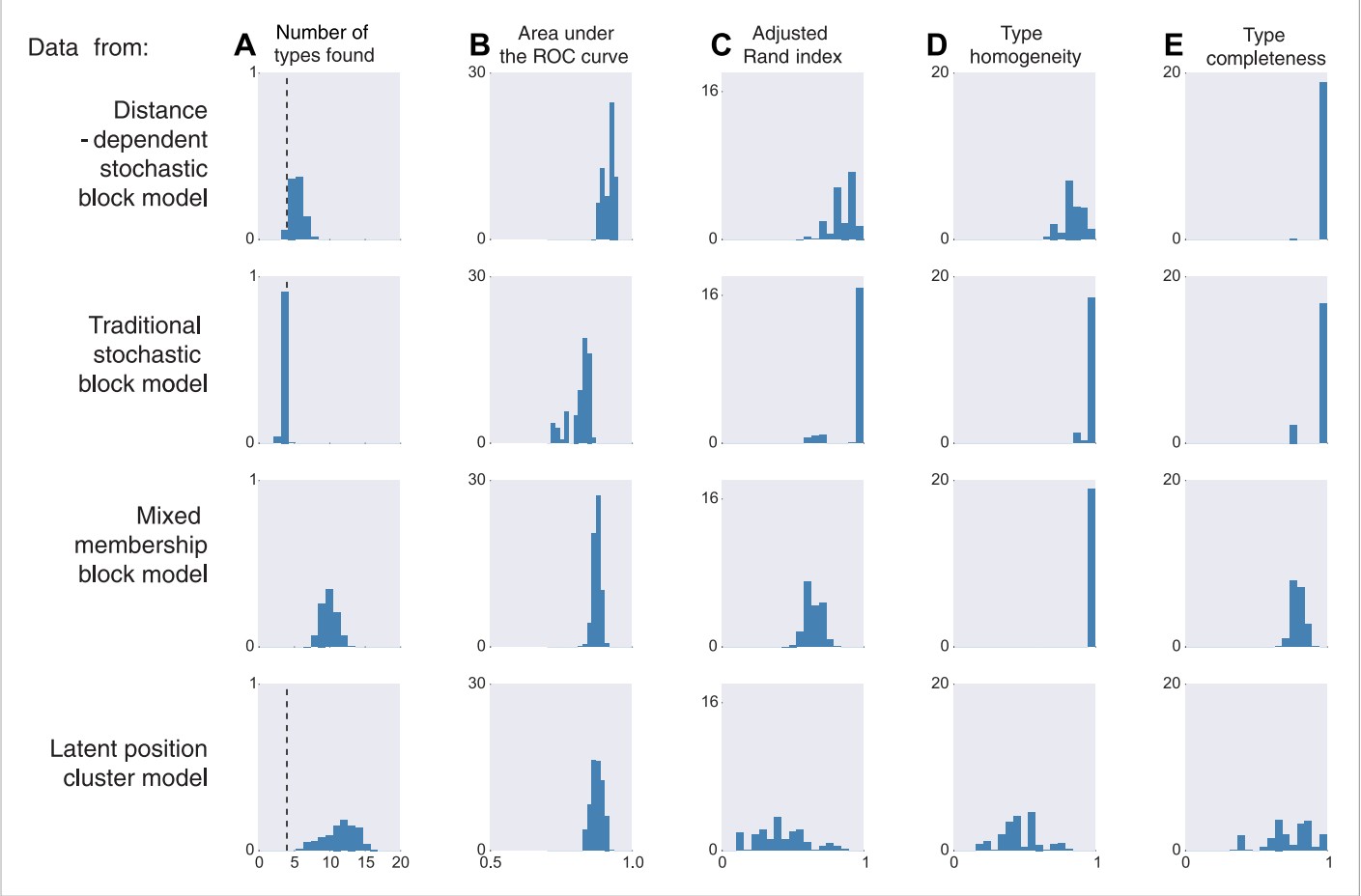

**Figure 3**. Model inferences when the true generating model differs from our distance-block-model prior. Horizontal columns show results with synthetic data generated according to the distance-dependent stochastic block model, the non-distance-dependent stochastic block model, the mixed membership block model, and the latent position cluster model. In all cases histograms represent posterior distribution over the indicated metric. (**A**) The number of types found by the model; the vertical dashed line indicates the 'true' type number (not applicable to the mixed membership model). (**B**) The area under the receiver operating characteristic (ROC) curve, indicating link prediction accuracy. (**C**, **D**, **E**) Clustering metrics quantifying degree of type agreement with known ground truth.

elimination of connectivity along the boundary might give the appearance of distance-dependent connectivity when there is none. We thus performed simulations to check if edge effects could destroy spatial structure and if edge effects could introduce artificial, spurious spatial structure. We measure the degree to which distance-dependent effects can arise from selecting regions that are smaller than the 'scale' of connectivity (*Figure 4*). We do this by generating two collections of synthetic datasets—one with distance-dependent connectivity and one without. We then in each dataset randomly examine contiguous circular regions with area varying from zero to the entire volume, and empirically calculate the spatial variance in type-dependent connectivity. We find that, if there is no distance dependence, edge effects do not artificially introduce distance dependence. However, if the section we are examining is too small, our model can miss the distance dependence. Thus with respect to distance-dependent connectivity inference, our model errs on the side of caution. But we also find that for spatial extent that is similar to the currently available datasets, the effects of this are quite limited.

## Learning types and circuitry in the retina

The mouse retina (*Masland, 2001*) is a neural circuit which we expect to have connectivity patterns that are well approximated by our generative model. It is known that there are multiple classes of cells

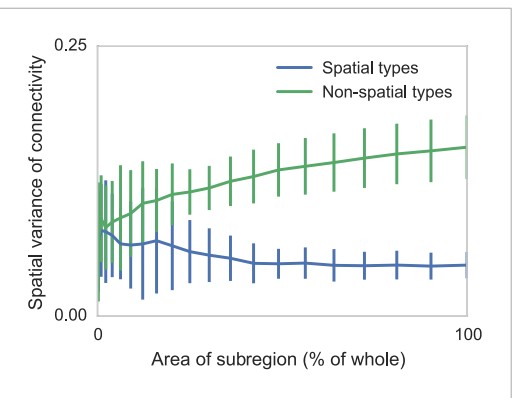

**Figure 4**. Two sets of generated synthetic data, one with spatially dependent connectivity and one without. We measure the variance in the connectivity-distance plot for randomly selected regions of each dataset, ranging from single cells to the entire volume. We see that while selecting too small a region can destroy the appearance of distance-dependent connectivity, it does not create it in non-spatial data.

that can be broadly grouped into: ganglion cells that transmit information to the rest of the brain; bipolar cells that connect between different cells; and amacrine cells that feed into the ganglion cells. Recent research (*Helmstaedter et al., 2013*) has produced a large dataset containing both the types of cells from orthogonal approaches, and also the connectivity matrix between all reconstructed cells (*Figure 5A*).

The algorithm took 8 hr to perform inference, dividing neurons into a set of cell types which reflect known neuroanatomical distinctions (*Figure 5* shows the MAP result). For each pair of neurons there is a specific distance-dependent connection probability (*Figure 5D*), which is well approximated by the model fit. Moreover, each type of cell is rather isotropically distributed across space (*Figure 5C*) as should be expected for true cell types.

Comparing the results of the algorithm to other information sources allows evaluation of the quality of the type determination. Our types closely reflect the (anatomist-determined) segmentation of cells into retinal ganglion, narrow amacrine, medium/wide amacrine, and bipolar cells (*Figure 6B*). We find that the types we find tend to reflect the known laminar distribution in the retina (*Figure 6C*) as well as the known synaptic density profiles.

The algorithm yields a separation of neurons into a smaller number of types than the fully granular listing of 71 types found by the original authors of the paper (*Helmstaedter et al., 2013*), although it is still highly correlated with those finer type distinctions (see section 'Mouse retina'). It is our expectation that, with larger datasets, even closer agreement would be found.

Our fully Bayesian model produces a distribution over probable clusterings. *Figure 6* shows this posterior distribution as a cell–cell coassignment matrix, sorted to find maximum block structure. Each large, dark block represents a collection of cells believed with strong probability to be of the same type. When we plot (*Figure 6B*) the anatomist-derived cell types along the left, we can see that each block consists of a roughly homogeneous collection of types.

We evaluate our model along three sets of parameters (*Figure 6*): how closely does our clustering agree with neuroanatomists' knowledge? Given two cells, how accurately can our model predict the link between them? And how closely does the spatial extent (within a layer) of our identified types agree with the spatial extent of types identified by neuroanatomists?

For our model we show the receiver operating characteristic (ROC) curve (*Figure 6D*) which shows how the true and false positive rates trade off. We plot the posterior distribution of the area under this curve in *Figure 6E*. We then plot the posterior distribution for cluster agreement metrics—completeness, homogeneity, and adjusted Rand index (ARI) (*Figure 6F*). We see that our model tends to over-cluster—cells which are of distinct type (at the finest granularity of neuroanatomist-identified type) are grouped as a single type by our model.

We compare link-prediction accuracy across the methods, including our own (*Figure 6G*, AUC, red). We find that given the dataset, many techniques allow for good link-predictive accuracy. All the methods allow decent link prediction with an AUC in the 0.9 range. However, our algorithm clearly outperforms the simple statistical models that only use connectivity.

As a second measure we compare link-prediction accuracy across the methods (*Figure 6G*, ARI, blue). We find that our algorithm far outperforms the controls. We also find that when it is based on more of the same information used by anatomists, then it gets better at agreeing with these anatomists. In particular, using connectivity, distance, synapse distribution, and soma depth leads to the highest ARI. When using the available information, the algorithm produces a good fit to human anatomist judgments.

Finally we look at the spatial extent of the discovered types both within a layer and between layers (*Figure 6H*). We see that, in the absence of distance information, mere connectivity information

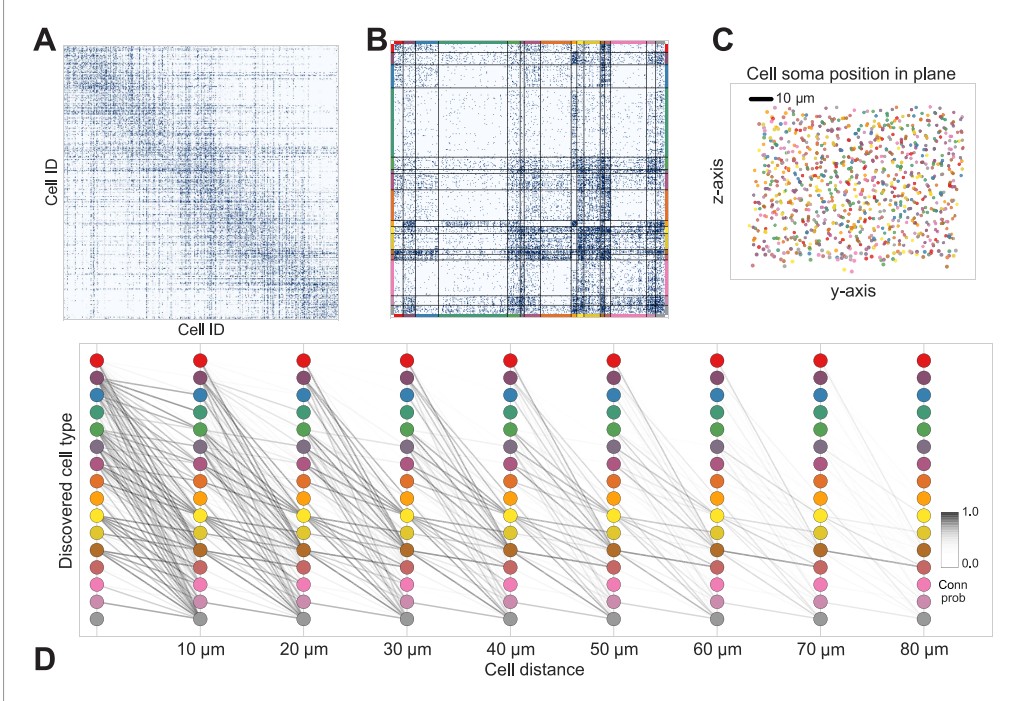

**Figure 5**. Discovering cell classes in the mouse retina connectome. Here we show the maximum a posteriori (MAP) estimate for the types in the mouse retina data. (**A**) Input connectivity data for 950 cells for which soma positions were known. (**B**) Clustered connectivity matrix; each arbitrary color corresponds to a single type and will be used to identify that type in the remainder of the plot. (**C**) The spatial distribution of our cell types—each cell type tessellates space. Colors correspond to those in (**B**). (**D**) Connectivity between our clusters as a function of distance—the cluster consisting primarily of retinal ganglion cells (brown nodes on the graph) exhibits the expected near and far connectivity. Conn prob: probability of connection.

results in types which only span a small region of space—essentially local cliques. Incorporation of distance information results in types which span the entire extent of the layer. The depth variance of all models continues to be substantially larger than that predicted by human anatomists—future directions of work include attempting to more strongly encode this prior belief of laminarity.

## Recovering spatial connectivity in multiple graphs simultaneously

Having shown our model to work on the repeating tessellated, laminar structure of the mammalian retina, we then apply our model to a structurally very different connectome—the whole body of a small roundworm: *C. elegans* is a model system in developmental neuroscience (*White et al., 1986*), with the location and connectivity of each of 302 neurons developmentally determined, leading to early measurement of the connectome. Unlike the retina, only the motor neurons in *C. elegans* exhibit regular distribution in space—along the body axis. Most interneurons are concentrated in various ganglia that project throughout the entire animal, and the sensory neurons are primarily located in a small number of anterior ganglia. *C. elegans* also differs from the retina in that the measured connectome is actually two separate graphs—one of directed chemical synapses and another of undirected electrical synapses. As this is a very different connectome, it allows an interesting generalization test: how well will our model work on such a distinct dataset?

Using both the chemical and electrical connectivity (see 'Materials and methods'), we determined the underlying cell types explained by connectivity and distance (*Figure 7A*). A superficial inspection of the results shows clustering into groups consisting roughly homogeneously of motor neurons, sensory neurons, and interneurons. Closer examination reveals agreement with the classifications originally outlined by White in 1986 (*White et al., 1986*).

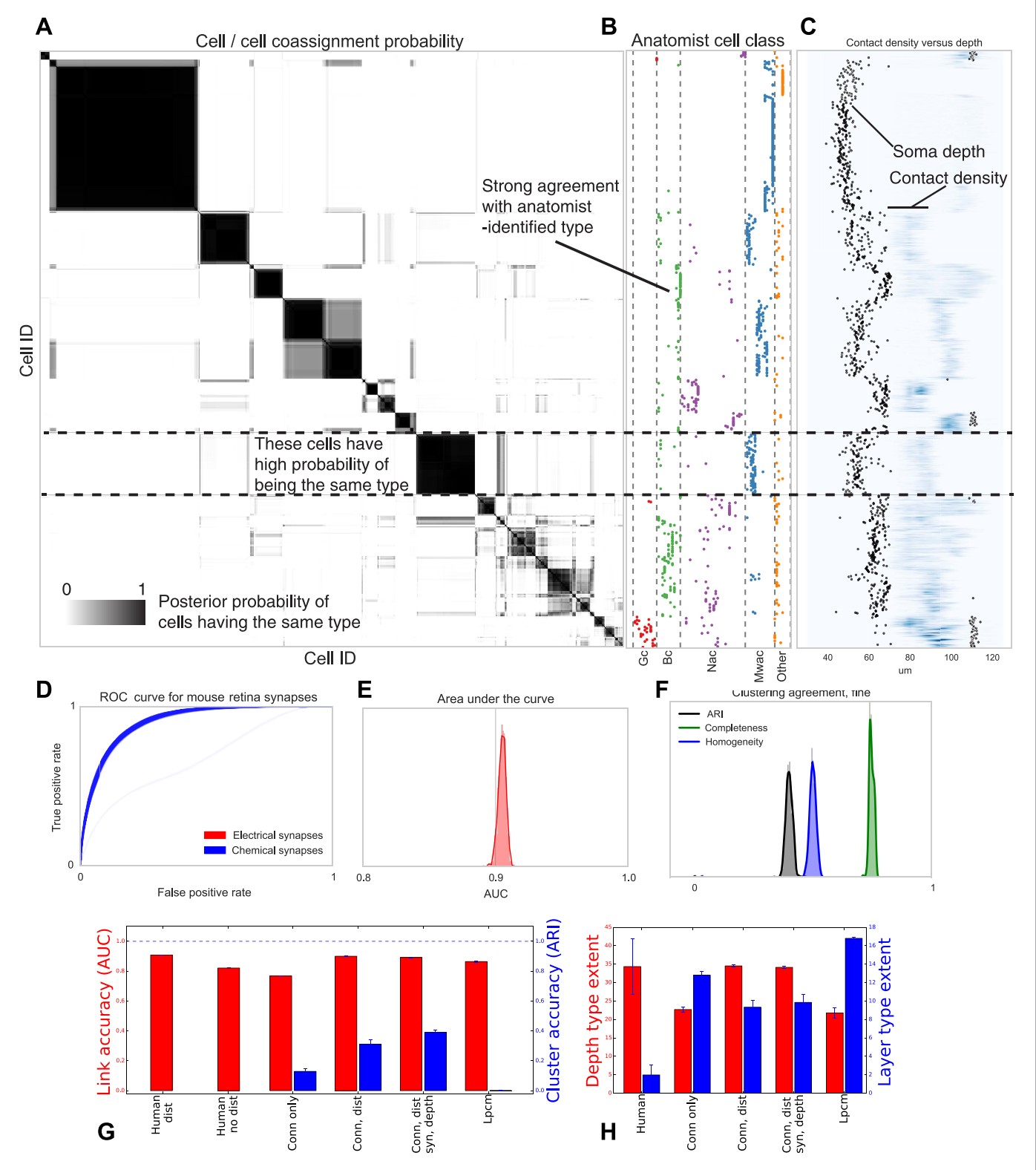

**Figure 6**. Visualizing type inference uncertainty. Our fully Bayesian model gives a confidence estimate (posterior probability) that any two given cells are of the same type. In (**A**) we visualize that cell–cell coassignment matrix, showing the probability that cell i is of the same type as cell j on a range from 0.0 to 1.0. The block structure shows subsets of cells which are believed to all belong to the same type. For comparison, (**B**) shows the anatomist-defined type for each cell, grouped broadly into the coarse types identified in the previous panel. (**C**) Link versus cluster accuracy. (**D**) The posterior distribution of receiver
*Figure 6. continued on next page*

*Figure 6. Continued*

operating characteristic (ROC) curves from 10-fold cross-validation when predicting connectivity, as well as (**E**) the area under the curve (AUC) and (**F**) the type agreements with known neuroanatomist types. ARI: adjusted Rand index. Model comparison, showing using human-discovered types with and without distance information, as well as our model incorporating just connectivity, connectivity and distance, or connectivity, distance, and synaptic depth (as well as the alternative latent position cluster model, see text). (**G**) A comparison of the predictive accuracy (AUC) for hand-labeled anatomical data, versus inclusion of additional sources of information, as well as the clustering accuracy. Note that our model sacrifices very little predictive accuracy for additional clustering accuracy. By comparison, conventional methods fail at one or both. ARI: adjusted Rand index. (**H**) The spatial extent (in depth and area) of the types identified by humans and our various algorithmic approaches.

Note our clustering does not perfectly reflect known divisions—several combinations of head and sensory neurons are combined, and a difficult-to-explain group of mostly VB and DB motor neuron types, with VC split between various groups. Our identified cell types thus reflect a 'coarsening' of known types, based entirely on connectivity and distance information, even when the organism exhibits substantially less spatial regularity than the retina.

## Types and connectivity in artificial structures

To show the applicability of our method to other connectome-style datasets, we obtained the spatial location and interconnectivity of the transistors in a classic microprocessor, the MOS Technology 6502 (used in the Apple II) (*James et al., 2010*). Computer architects use common patterns of transistors when designing circuits, with each transistor having a 'type' in the circuit. We identified a region of the processor with complex but known structure containing the primary 8-bit registers X, Y, and S (*Figure 8*).

Our algorithm identifies areas of spatial homogeneity that mirror the known structure in the underlying architecture of the circuit, segmenting transistor types recognizable to computer architects. Using the original schematics, we see that one identified type contains the 'clocked' transistors, which retain digital state. Two other types contain transistors with pins C1 or C2 connected to ground, mostly serving as inverters. An additional identified type controls the behavior of the three registers of interest (X, Y, and S) with respect to the SB data bus, either allowing them to latch or drive data from the bus. The repeat patterns of spatial connectivity are visible in *Figure 8C*, showing the man-made horizontal and vertical layout of the same types of transistors.

## Discussion

We have presented a machine learning technique that allows cell types and microcircuitry to be discovered from connectomics data. We have shown its applicability to regularly structured laminar neural circuits like the retina, as well as a less structured whole neuronal organism (*C. elegans*) and a classic processor. When compared to existing methods, we show how the incorporation of all of this data yields results that combine both high link-prediction accuracy and high agreement with human anatomists. We have found that combining the available data types allows us to discover cell types and microcircuitry that were known to exist in the systems based on decades of previous research and allows good prediction of connectivity.

For our probabilistic models, no known solution exists to exactly find the most probable parsing of the neurons into cell types and connectivity patterns. We employ a collection of MCMC techniques (see 'Materials and methods'), but while different initializations converge to similar ultimate values, we can never realistically obtain the global optimum. There are a broad range of techniques that may offer good approximations to the global optimum and future work could adapt them to find more precise solutions to our problem.

For our probabilistic model, inference becomes slower as the amount of data increases. Our algorithm required several hours for 1000 neurons. Scaling this class of probabilistic model is an active area of research, and recent results in both variational methods (*Hoffman et al., 2013*) and spectral learning (*Anandkumar et al., 2012*) and future work could adapt them to find faster approximate solutions to our problem.

Larger datasets will allow algorithms to distinguish more distinct types and we expect closer agreement with existing anatomical knowledge as more data become available. Moreover, in general, for such problems precision increases with the size of the dataset and the cells that we have are not sufficient to statistically distinguish all the cell types known in anatomy (such as the ~70 in the retina). Still, using only connectivity and distance, it is possible to meaningfully divide neurons into types.

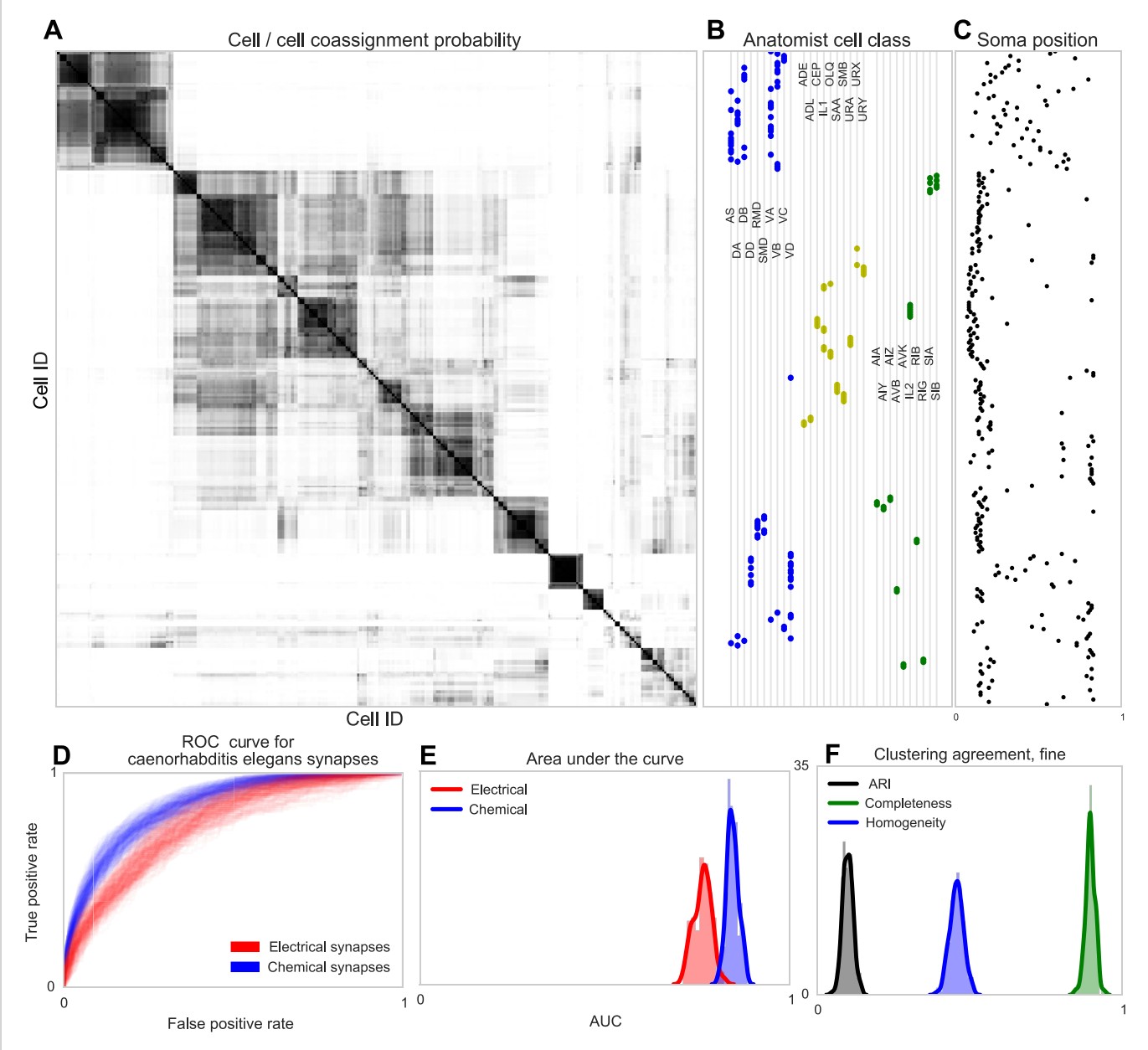

**Figure 7**. Discovering connectivity and type in *C. elegans*. (**A**) Posterior distribution on cell connectivity as a function of discovered type, similar to *Figure 6*. In (**B**) we plot neuroanatomist-derived types along with their labels. Our model shows a high probability of motor neurons, sensory neurons, and various interneuron classes being of the same type. Soma positions along the body axis are plotted in (**C**) where we see that we cluster spatially distributed motor neurons together, whereas head sensory neurons are more likely to be grouped together as well. (**D**) The receiver operating characteristic (ROC) curves for held-out link probability for both the electrical synapses (gap junctions) and chemical synapses in *C. elegans*. (**E**) The posterior distribution of the area under the ROC curve (AUC) for the curves in (**D**). (**F**) Measurements of the agreement of our identified cell types compared to neuroanatomists. The high completeness but low homogeneity (and corresponding low adjusted Rand index, ARI) reflects our model's tendency to group multiple types into a single type.

Our small collection of hand-selected distance-dependent likelihood functions is clearly non-exhaustive, and assumes monotonicity of connectivity probability—for a given class, closer cells are never less likely to connect. This is known to be insufficient for various neural systems. Future models could incorporate a wider variety of likelihood functions, or even learn the global functional form from the data.

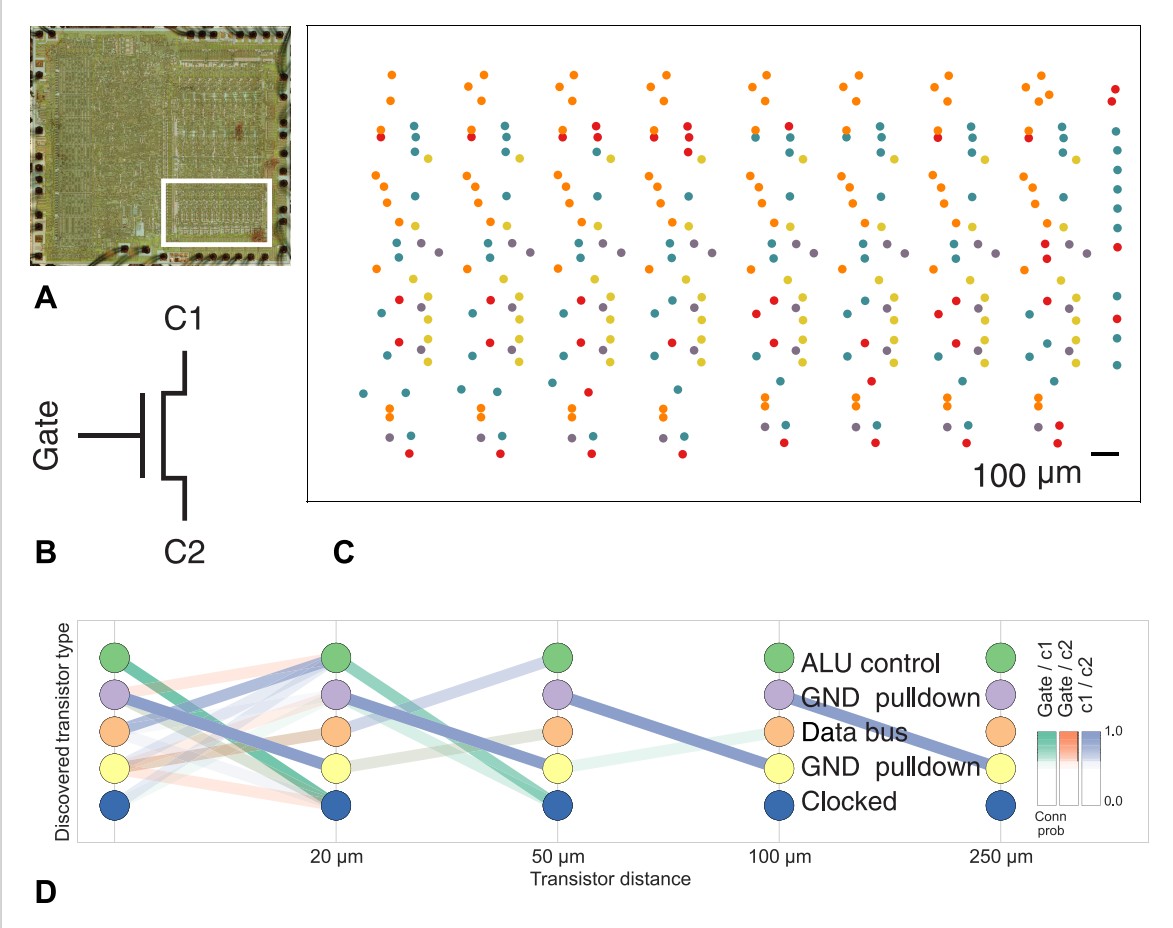

**Figure 8**. Discovering connectivity and type in the MOS 6502 microprocessor. (**A**) The micrograph of the original microprocessor, with the region containing the registers under study highlighted. (**B**) Our graph consists of the interconnections of MOS field-effect transistors with three terminals, Gate, C1, and C2. The reconstruction technique did not permit resolution of C1 and C2 into source and drain. (**C**) The spatial distribution of the transistors in each cluster show a clear pattern. (**D**) The clusters and connectivity versus distance for connections between Gate and C1, Gate and C2, and C1 and C2 terminals on a transistor. Purple and yellow types have a terminal pulled down to ground and mostly function as inverters. The blue types are clocked, stateful transistors, green control the ALU and orange control the special data bus (SDB).

There are a range of previous approaches to the discovery of neural microcircuitry (*Mountcastle, 1957*; *Douglas and Martin, 1991*; *Freund and Buzsáki, 1998*; *Barthó et al., 2004*). These generally involve a great deal of manual labor and ad hoc determination of what constitutes a type of cell—to this day there are disagreements in the literature as to the true types in the mammalian retina. Much as phylogenomics has changed our understanding of animal ontologies, modern large scale data will allow the efficient unbiased discovery of cell types and circuits. The sheer amount of available data demands the introduction of algorithmic approaches.

The development of automatic identification and quantification of cell type may also provide a new computational phenotype for quantifying the effect of disease, genetic interventions, and developmentally experienced neural activity. Our method can in principle identify neuron types across non-connected graphs, for example, across animals. For example, the types of neurons in one animal can be associated with the types of neurons in another animal, in the same way as this is already possible through molecular markers (*Brown and Hestrin, 2009*). This could be particularly important if cell types appear that are due to properties of the stimuli and experience as opposed to just the molecular properties of cells, such as color and orientation selective types in primary visual cortex (*Lennie and Movshon, 2005*; *Sincich and Horton, 2005*). This would allow comparative quantitative anatomy across animals, and aid the search for the ultimate causes of connectivity.

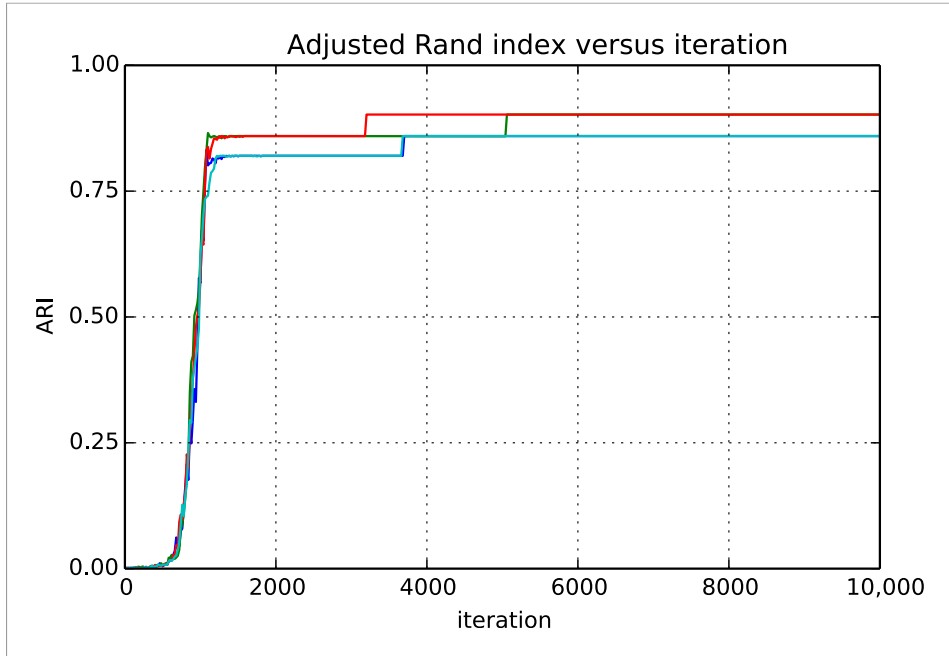

**Figure 9**. Adjusted Rand index (ARI) for synthetic data as a function of run iteration.

Our model combines connectivity, cellular and synaptic properties, and suggests the way towards combining even richer data. Distinct cell types differ in morphology, connectivity, transcriptomics, relation to behavior or stimuli, and many other ways. Algorithms combining these data and type information may allow us to synthesize all the available information from one experiment or even across experiments into a joint model of brain structure and function.

Our work shows how rich probabilistic models can contribute to computational neuroanatomy. Eventually, algorithms will have to become a central tool for anatomists, as it will progressively become impossible for humans to parse the huge datasets. This transition may follow a similar transition to that of molecular biology (with gene-finding algorithms) and evolutionary biology (with computational phylogenetics). Ultimately, computational approaches may help resolve the significant disagreements across human anatomists.

## Methods

### Probabilistic model

Our model is a extension of the iSBM (*Kemp et al., 2006*; *Xu et al., 2006*) to incorporate spatial relations between entities, inspired by attempts to extend these models with arbitrary discriminative functions (*Murphy, 2012*).

We take as input a connectivity matrix $R$ defining the connections between cell $e_i$ and $e_j$, as well as a distance function $d(e_i, e_j)$ representing a (physical) distance between adjacent cells. See the supplemental material for extension to multiple connectivity matrices. We assume there exist an unknown number $K$ of latent (unobserved) cell types, $k \in \{1, 2, 3, \ldots, K\}$, and that each cell $e_i$ belongs to a single cell type. We indicate a cell $e_i$ is of type $k$ using the assignment vector ($c$), so $c_i = k$. The observed connectivity between two cells $R(e_i, e_j)$ then depends only on their latent type and their distance through a link function $f(\cdot, d(e_i, e_j))$. We assume $f$ is parameterized based on the latent type, $c_i = m$ and $c_j = n$, via a parameter $\eta_{mn}$, as well as a set of global hyperparameters $\theta$, such that the link function is $f(d(e_i, e_j)|\eta_{mn}, \theta)$.

We then jointly infer the MAP estimate of the class assignment vector ($c$) = $\{c_i\}$, the parameter matrix $\eta_{mn}$, and the global model hyperparameters $\theta$:

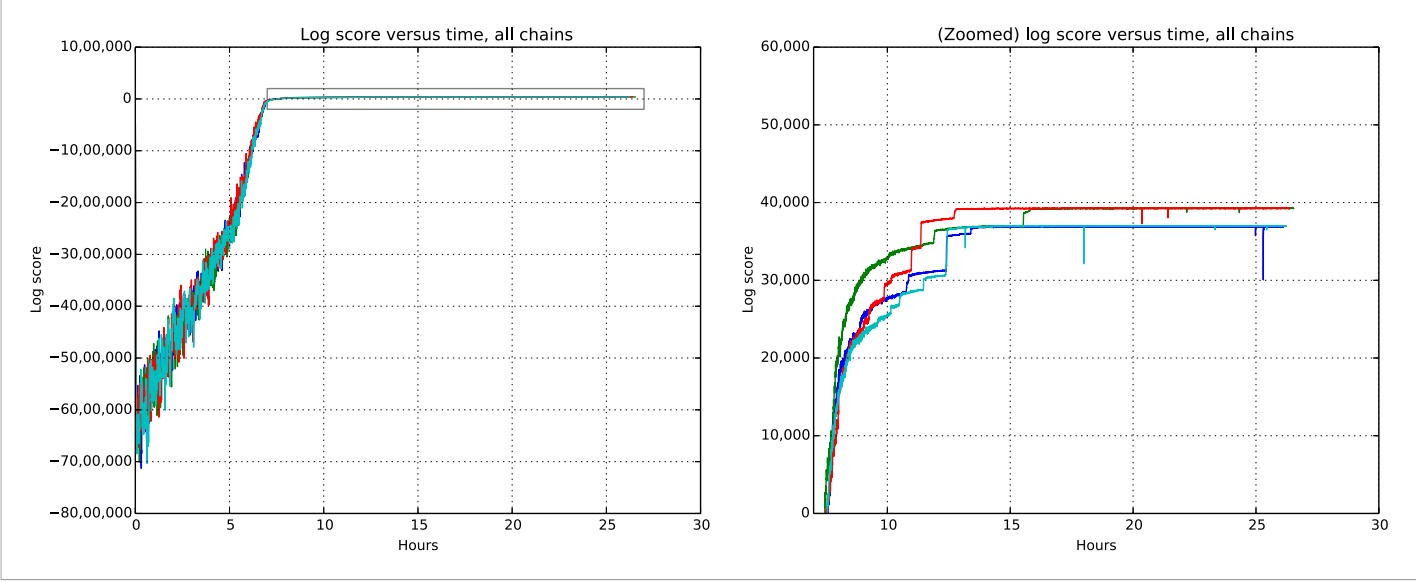

**Figure 10**. Total model score (log score) versus wall clock time.

$$p(\mathbf{c}, \eta, \theta | R) \propto \prod_{i,j} p\Big(R(e_i, e_j) | f\Big(d(e_i, e_j) | \eta_{c_i c_j}\Big), \theta\Big) \prod_{m,n} p(\eta_{mn} | \theta) p(\theta) p(\mathbf{c} | \alpha) p(\alpha) p(\theta). \tag{2}$$

We describe the spatial 'logistic-distance Bernoulli' function here, and others in the supplemental material.

The 'logistic-distance Bernoulli' spatial model assumes that, if cell $e_i$ is of type $m$ and cell $e_j$ is of type $n$, then $\eta_{mn} = (\mu_{mn}, \lambda_{mn})$, and the probability that two cells $e_i$ and $e_j$ are connected is given by

$$p^* = \frac{1.0}{1 + \exp \frac{d(e_i, e_j) - \mu_{mn}}{\lambda_{mn}}}, \tag{3}$$

$$p = p^* \cdot (p_{max} - p_{min}) + p_{min}, \tag{4}$$

where $p_{max}$ and $p_{min}$ are global per-graph parameters.

We place exponential priors on the latent parameters:

$$\mu_{mn} \sim \exp\Big(\mu | \mu^{hp}\Big), \tag{5}$$

$$\lambda_{mn} \sim \exp\Big(\lambda | \lambda^{hp}\Big), \tag{6}$$

using $\lambda^{hp}$ and $\mu^{hp}$ as global per-graph hyperparameters.

We use a Dirichlet-process prior on class assignments, which allows the number of classes to be determined automatically. In brief, for $N$ total cells, the probability of a cell belonging to a class is proportional to the number of data points already in that class, $N_k$, such that $p(c_i = k) \propto \frac{m_k}{N + \alpha}$ and the probability of the cell belonging to a new class $k'$ is $p(c_i = k') \propto \frac{\alpha}{N + \alpha}$. $\alpha$ is the global concentration parameter—larger values of $\alpha$ make the model more likely to propose new classes. We grid the parameter $\alpha$ and allow the best value to be learned from the data.

Where we model cell depth, we assume that each cell type has a typical depth, and thus a Gaussian distribution of $s_i$. We assume $s_i \sim N(\mu_k^{(s)}, \sigma_k^{2(s)})$, where the (s) superscript indicates these model parameters are associated with the soma-depth portion of our model. We use a conjugate prior for $(\mu_k^{(s)}, \sigma_k^{2(s)})$ with $\mu_k^{(s)} \sim N(\mu_{hp}^{(s)}, \sigma_k^{2(s)}/\kappa_{hp}^{(s)})$ and $\sigma_k^{2(s)} \sim \chi^{-1}(\sigma_{hp}^{2(s)}, \nu_{hp}^{(s)})$. The use of conjugacy simplifies inference while allowing for each cell type to have its own depth mean and distribution.

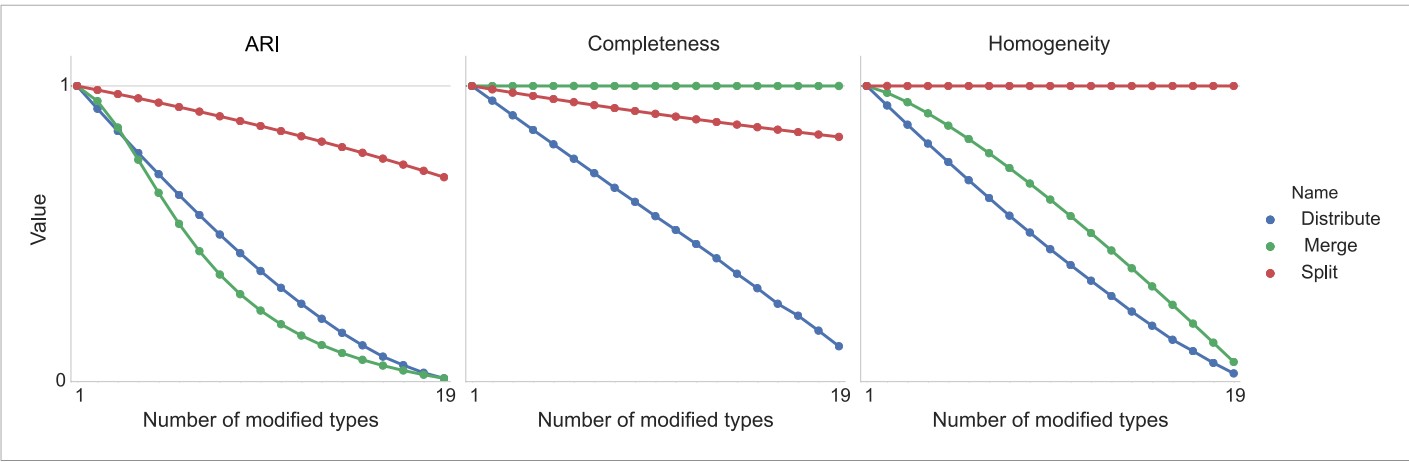

**Figure 11**. Type agreement evaluation metrics as a function of splitting types, merging types, and randomly distributing cells between types.

We model synapse depth profile that each cell type has a characteristic depth distribution of synaptic contact points, and mixture of Gaussian distributions over cell is $N_i$ contact points, $\mathbf{g}^i$. We do this by assuming the $g_j^i$ are drawn from an $M = 3$-component mixture of Gaussians. Thus associated with each cell type $k$ is a vector of $M$ Gaussian means $(\mu_{k,1}^g, \cdots, \mu_{k,M}^g)$, and a mixture vector $\pi_k$. This representation can thus model depth distributions of contact points that have up to three modes, an assumption that is well matched in the bulk of anatomical studies of cell-type-dependent connectivity.

## Inference

We perform posterior inference via MCMC, annealing on the global likelihood during the traditional burn-in phase. MCMC transition kernels for different parts of the state space can be chained together to construct a kernel whose ergodic distribution is the target ergodic distribution over the entire state space.

Our first transition kernel ('structural') performs Gibbs sampling of the assignment vector $p(\mathbf{c}|\eta, \theta, \alpha)$. The lack of conjugacy in our likelihood model makes an explicit evaluation of the conditional assignment probabilities impossible, motivating us to use an auxiliary variable method (**Neal, 2000**) in which a collection of ephemeral classes is explicitly represented for the duration of the Gibbs scan.

We then employ a transition kernel to update the per-component parameter values $\eta_{mn}$. Conditioned on the assignment vector $\mathbf{c}$ and the model hyperparameters $\theta$, $\alpha$ the individual $\eta_{mn}$ are independent. We slice sample (**Neal, 2003**) each component's parameters, choosing the slice width as a function of the global hyperparameter range.

The global hyperparameters, both $\alpha$ and $\theta$, are allowed to take on a discrete set of possible values. As $\theta$ is often a tuple of possible values, we explore the Cartesian product of all possible values. We then Gibbs sample (our final transition kernel), which is always possible in a small, finite, discrete state space.

We chain these three kernels together, and then globally anneal on the likelihood from a temperature of $T = 64$ down to $T = 1$ over 900 iterations unless otherwise indicated, and then run the chain for another 100 iterations. We then generate at least 20 samples, each taken from the end of a single Markov chain initialized from different random initial points in the state space. For visualization we pick the chain with the highest log likelihood, but for all numerical comparisons (including link probability and cluster accuracy) we use this full collection of samples from the posterior distribution to estimate the resulting statistics.

## Link prediction

To compute link-prediction accuracy, we compute the probability of a link between two cells using each model, trained via 10-fold cross-validation. We use a full collection of posterior samples when computing the link probability, and then compute the area under the ROC curve for each.

We compare our model with a standard network clustering model, the latent-position clustering model. This model assumes each cell belongs to one of $K$ clusters, and each cluster is associated with a $d$-dimensional Gaussian distribution. The probability of a link is then a function

of the distance between the data points in this continuous space. We use a variational implementation provided in R (*Salter-Townshend and Murphy, 2013*), parametrically varying the number of latent dimensions and the number of requested groups. While this model provides reasonable link-predictive accuracy, the clusterings dramatically disagree with those from human anatomists.

## Parameters

Hierarchical generative models can be sensitive to hyperparameter settings, thus for most hyperparameters we perform inference. In cases where we cannot, we run separate collections of Markov chains at separate settings and show the results across all pooled parameters. For the case of the mouse retina data, we consider maximum link probability $p_{max} \in \{0.95, 0.9, 0.7\}$, variance scales for the synapse density profile of $\sigma^2 \in \{0.01, 0.1, 1.0\}$ (of normalized depth), and $K \in \{2, 3\}$ possible synapse density profile mixture components. For the connectivity-distance-only model we actually perform inference over both $p_{max}$ and $p_{min}$.

## Mixing of our Markov chains

Evaluating whether or not approximate inference methods, such as MCMC, produce samples which are valid approximations of the posterior distribution is an ongoing area of research in the computational statistics community. We use a rough proxy here—synthetic likelihood evaluation. For synthetic datasets of sizes comparable to our real data size, do we recover known ground truth information after running our Markov chains for the appropriate amount of time?

Figures 9 and 10 show the cluster accuracy (ARI) to ground truth and the total log score as a function of runtime. We see dramatic changes in log score initially as we vary the temperature, stabilizing as runtime progresses, for each chain. Then we see the characteristic jumps between nearby modes towards the end of the run, in both log score and ARI. Importantly, regardless of whether our model over- or under-estimates the exact posterior variance about the network, we find points in the latent variable space that are both predictive *and* parsimonious, largely agreeing with the human anatomists and predicting existing connections.

## Dataset details

### Mouse retina

Dense serial electron microscopy of a 114 μm × 80 μm area in the mouse retina by *Helmstaedter et al. (2013)* yielded a listing of places where neurons come into contact. There were over 1000 cells originally, and we selected the 950 for which the location of the soma could be reconstructed from the provided cell plots (soma locations were not provided by the study's authors in machine-readable form). The result was a matrix of the total synapse-like contact area between all pairs of 950 cells. Area was thresholded at 0.1 μm, determined by hand, to yield a 950 × 950 entry matrix that served as input to our algorithm. We measured the distance between cells using the reconstructed soma centers, and used the logistic-distance spatial relation. Hyperprior griddings are shown in the 'Hyperprior grids and hyperprior inference' section.

### C. elegans

We obtained the connectome of *C. elegans* from data published previously (*Varshney et al., 2011*), and isolated the 279 non-pharyngeal neurons, with a total of 6393 chemical synapses and 890 gap junctions originally cleaned up in *Chen et al. (2006)*. A cell's position was its distance along the anterior–posterior axis normalized between 0 and 1. We used both networks, the chemical network as a directed graph and the electrical network as an undirected graph. We use the synapse counts with the logistic-distance Poisson likelihood, scaling the counts by 4.0 to compensate for the Poisson's overdispersion.

### Microprocessor

We extracted the connection graph for the transistors in the MOS 6502 (*James et al., 2010*). Each transistor has three terminals (gate, source, drain), but the methods of the original dataset were unable to consistently resolve which of the C1 and C2 terminals were source and drain, leading to ambiguity in our encoding. We identified a region consisting of three registers X, Y, and S via visual inspection and focused our efforts there. We created a total of six connectivity matrices by examining possible terminal pairings. For example, one graph encodes the connectivity between pins g and $c_1$. We then have, $R^{gc_1}(e_i, e_j) = 1$ if transistor $e_j$ and $e_j$ are connected via pins $g$ and $c_1$.

## Other likelihoods

We reparameterized the logistic-distance Bernoulli likelihood to better capture the microprocessor data structure. We are explicitly setting the maximum probability $p$ of the logistic function on a per-component basis, drawing from a global $p \sim \text{Beta}(\alpha_{hp}, \beta_{hp})$. Then $\lambda$ is set for each component as a global hyperparameter, $\lambda$.

The 'logistic-distance Poisson' spatial model is used to explicitly model the count of synapses, $c$, between two neurons. The probability of $c$ synapses between two neurons is distributed $c \sim \text{Poisson}(c|r)$, where $r$ (the 'rate') is generated by a scaled logistic function (the logistic function has range [0, 1]). For each component $\eta_{mn}$ we learn both the threshold $\mu_{mn}$ and the rate scaling factor $r_{mn}$. Thus if cells $m$ and $n$ are likely to have on average 20 synapses if they are closer than 5 μm, then $\mu_{mn} = 5$ and $r_{mn} = 20$.

Thus the probability of $R(e_i, e_j) = c$ synapses between two cells $e_i$ and $e_j$ is given by:

$$r^* = \frac{1.0}{1 + \exp \frac{d(e_i, e_j) - \mu_{mn}}{\lambda}}, \tag{7}$$

$$r = r^* \cdot (r_{mn} - r_{min}) + r_{min}, \tag{8}$$

$$R(e_i, e_j) \sim \text{Poisson}(c|r), \tag{9}$$

where $\lambda$ and $r_{min}$ are per-graph parameters and we have per-component parameters $\mu_{mn} \sim \text{Exp}(\mu|\mu^{hp})$ and $r_{mn} \sim \text{Exp}(r_{mn}|r_{scale}^{hp})$.

## Source code and data

All source code and materials for running experiments can be obtained from the project website, at http://ericmjonas.github.io/connectodiscovery/.

All preprocessed data has been made publically available as well.

## Extension to multiple graphs

The model can handle multiple graphs $R^q$ simultaneously with a shared clustering by extending the likelihood to include the product of the likelihoods of the individual graphs.

$$p(\mathbf{c}, \eta^q, \theta^q | R^q) \propto \prod_q \left( \prod_{i,j} p(R^q(e_i, e_j) | f(d(e_i, e_j) | \eta_{c_i c_j}^q, \theta^q) \prod_{m,n} p(\eta_{mn}^q | \theta^q) p(\theta^q) \right) p(\mathbf{c}|\alpha) p(\alpha). \tag{10}$$

## Hyperprior grids and hyperprior inference

For the mouse retina logistic-distance Bernoulli model, we gridded $\mu^{hp}$ and $\lambda^{hp}$ into 40 $\log_{10}$-spaced points 1.0 and 80.

For the *C. elegans* data with the logistic-distance Poisson model, we gridded $\mu_{hp}$ and $\lambda$ into 20 $\log_{10}$-spaced points between 0.2 and 2.0, and the *ratescale*$^{hp}$ parameter into 20 $\log_{10}$-spaced points between 2.0 and 20.0. We globally set *rate*$_{min}$ = 0.01.

For the microprocessor with the logistic-distance with fixed lambda parameter and Bernoulli likelihood, we gridded $mu_{hp}$ into 50 $\log_{10}$-spaced points between 10 and 500 and set $\lambda = \mu_{hp}/10$. $p_{min} \in \{0.001, 0.01, 0.02\}$ and both $p_\alpha$ and $p_\beta \in \{0.1, 1.0, 2.0\}$.

## Measuring clustering similarity

We compare discovered types to known types via cluster comparison metrics: cluster homogeneity, cluster completeness, and the ARI. Homogeneity measures how many true types are in a given found type. If every cell type is split into two types, each subtype is still completely homogeneous. Completeness measures how many members of a given true type are split across found types.

ARI takes into account both effects (*Hubert and Arabie, 1985*)—two identical clusterings have an ARI of 1.0, while progressively more dissimilar clusters have lower ARIs, becoming slightly negative as the clustering gets anti-correlated.

*Figure 11* shows the result of taking 20 different clusters and moving data points between them according to the following operations.

- **distribute**: take a class and distribute its elements uniformly among the remaining types.
- **merge**: take a type and merge it into another existing type.
- **split**: take a type and split it into two distinct types.

We can see the impact on ARI, completeness, and homogeneity as we perform these operations on more of the original 20 types. In all cases, 'distribution' of one type among the others is detrimental to the metric. Splitting impacts completeness but not homogeneity, and merging impacts homogeneity but not completeness.

## Acknowledgements

We thank Josh Vogelstein for discussions and reading of the manuscript, Finale Doshi-Velez for early discussions on the model, and Erica Peterson, Jonathan Glidden, and Yarden Katz for extensive manuscript review. Funding for compute time was provided by Amazon Web Services 'AWS in Education' grants.

## Additional information

### Funding

| Funder | Grant reference | Author |
|---|---|---|
| University of California Berkeley (University of California, Berkeley) | AWS in Education grant | Eric Jonas |
| National Science Foundation | NSF CISE Expeditions Award CCF-1139158 | |
| Lawrence Berkely National Laboratory | Award 7076018 | |
| Defense Advanced Research Projects Agency | XData Award FA8750-12-2-0331 | |
| National Institutes of Health | R01NS074044 | Konrad Kording |
| National Institutes of Health | R01NS063399 | Konrad Kording |

The funders had no role in study design, data collection and interpretation, or the decision to submit the work for publication.

### Author contributions

EJ, KK, Conception and design, Analysis and interpretation of data, Drafting or revising the article

## Additional files

### Major dataset

The following dataset was generated:

| Author(s) | Year | Dataset title | Dataset ID and/or URL | Database, license, and accessibility information |
|---|---|---|---|---|
| Jonas E | 2015 | Connectomics datasets | https://github.com/ericmjonas/circuitdata | Canonical, cleaned-up datasets, publicly available at GitHub, originally published in *Varshney et al., 2011* (http://dx.doi.org/10.1371/journal.pcbi.1001066) and *Helmstaedter et al., 2013* (http://dx.doi.org/10.1038/nature12346). |

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
