## [Decision Letter]

Thank you for sending your work entitled “Automatic discovery of cell types and microcircuitry from neural connectomics” for consideration at *eLife*. Your article has been evaluated by a Senior editor, a Reviewing editor and three reviewers, and it was decided that a revised submission should be considered.

The Reviewing editor and the reviewers discussed their comments before we reached this decision, and the Reviewing editor has assembled the following comments to help you prepare a revised submission.

Overall, the reviewers felt that the work potentially presents a major advance, addressing the problem of automatic analysis of large, augmented connectomic data sets, a problem that is quickly growing in importance as these data sets become available. As such, the work is considered to be very timely. However, it was also felt that it is unclear how the algorithm works and what its limitations might be. A more careful description and presentation of the method with added discussions is strongly needed. In other words, does the method actually work? As presented, it seems magical and impressive, and there was some concern as to whether it was too good to be true. Data and methods need to be made available as well as having controls and showing ROC curves to allow further evaluation. More rigor in defining terms and labeling figures is also needed. It was also felt that the figures, while aesthetically pleasing, were not helpful for evaluation of the algorithm.

Specifically, the following aspects were raised:

1) Explain the algorithm limitations, i.e., where it breaks down (noting that the reviewers appreciate that no algorithm can solve every problem). This might be best done in the context of a toy problem. Any potential user of this algorithm would benefit from understanding its limitations better. Under what conditions does it break down? What are the pressure points? For example, what if the data set has structure, but there really aren't distinct classes (as has been proposed for cortex)? Is it particularly sensitive to some of the choices of priors? Etc. Perhaps other examples would provide better insight into potential failure modes.

2) Show cross-validation on the link prediction.

3) As given, the cell type identity analysis is difficult to parse; the circos-style plots don't tell much and the other plots are often inadequately labelled. It is also unclear as to how validation was performed or was to be observed. Simply seeing ROC curves (or reported areas) or something similar is needed. The figures are of low informational value. How would some well-known supervised method do at this task in cross-validation? It would seem, intuitively, to be a natural choice for learning cell identity. See, for example, PMID: 21154911, for a well-defined analysis.

Show ROC style prediction on the cell types. This can be done by cross-validation i.e., do the clustering, then label all the cells using, e.g., 2/3 of the human-labelled data. Then assign each unknown cell a label based on its co-membership with cells of known labels. This would end up giving a ranked score, since probabilistic. The comparison method would be something like nearest-neighbor assignment. To avoid making this a multi-class prediction problem, one could do it one class at a time and not worry about overlapped assignments. This would set a good standard for methods comparison.

4) Consideration and discussion about the edge effects in the retinal data.

The retina data set covers a relatively small volume of the retina (on the order of 100 microns in each direction) and most of the cells whose cell bodies are located within this volume have their neurites “cut off”, i.e. they are not fully contained within the volume. This will lead to edge effects that may impact or bias connection patterns, for example cells whose cell bodies are near the center of the field of view should have more synaptic contacts (more partners in the contact matrix) simply because they are more fully captured in the reconstruction. How have such potential edge effects been accounted for in the present analyses? Could there be improvement in the performance of the clustering algorithm by taking into account such biases?

5) Chip example was considered less critical as compared to providing more testing and further details and discussion on the retinal and *C.elegans* examples.

6) The use of connectivity information for characterizing nodes in a neural network has a long history in cognitive and systems neuroscience. It would be nice to point this out. A potential starting point is the review of Passingham et al. (2002) in Nature Reviews Neuroscience.

7) Additional comments regarding unhelpful/unclear figure aspects for algorithm specifics:

Are there any known inherent biases in terms of preferences in cluster sizes? It appears, from Figure 4, that there are a small number of fairly large (and quite uniform) clusters and a larger number of quite small ones (that seem quite noisy with respect to the anatomist's cell classification).

It is difficult to judge just by visual inspection alone how “good” the agreement between the automated cluster analysis and the anatomist's ordering scheme really is (Figure 3, outer ring). Is there a more quantitative way of determining the “fit” of the model with the a priori anatomical cell type assignments?

In a similar vein, the text claims (in the subsection headed “Recovering spatial connectivity in multiple graphs simultaneously”) that for *C. elegans* the clusters correspond roughly homogenously to moto, sensory and interneurons, but looking at Figure 5, it is not so clear. Interneurons are found in almost all of the clusters, and very few are truly homogeneous with respect to these three types of neurons. Again, it would be good to think of a way to quantify “homogeneity”.

In the circular plots (e.g. Figure 1), it is not quite clear what the shaded arcs in the center refer to. Is the shading proportional to some sort of density (of what?), and what does the width of these arcs refer to?

---

## [Author Response]

*However, it was also felt that it is unclear how the algorithm works and what its limitations might be – a more careful description and presentation of the method with added discussions is strongly needed. In other words, does the method actually work? As presented, it seems magical and impressive, and there was some concern as to whether it was too good to be true. Data and methods need to be made available as well as having controls and showing ROC curves to allow further evaluation. More rigor in defining terms and labeling figures is also needed. It was also felt that the figures, while aesthetically pleasing, were not helpful for evaluation of the algorithm*.

We have added a broad range of additional detailed analysis, added additional figures to directly address the usefulness of the algorithm. We have also added simulations to show under which circumstances the algorithm fails. Indeed, the fact that the reviewers ask for an analysis of failure modes instead of stating (as some high profile journals) that any of the (unavoidable) failure modes would be reasons for rejection, truly impressed us.

*Specifically, the following aspects were raised*:

*1) Explain the algorithm limitations, i.e., where it breaks down (noting that the reviewers appreciate that no algorithm can solve every problem). This might be best done in the context of a toy problem. Any potential user of this algorithm would benefit from understanding its limitations better. Under what conditions does it break down? What are the pressure points? For example, what if the data set has structure, but there really aren't distinct classes (as has been proposed for cortex)? Is it particularly sensitive to some of the choices of priors? Etc. Perhaps other examples would provide better insight into potential failure modes*.

We have added simulations that show the failure modes of the algorithms. There are two kinds of failures: (1) the algorithm may not succeed at correctly solving the problem it is meant to solve, (2) the problem may not succeed at the problem we scientists want it to solve because it is really solving a different problem. We have added simulations to get at these two issues. In short, as we had already shown with respect to (1), the algorithm works worse and worse as the classes are made more and more similar as this slows down mixing. Such behavior is typical for MCMC algorithms. With respect to (2), we constructed cases where the algorithm infers cell-types where they are none, in the case of continuous distributions and cases where the algorithm does not find the cell-types because other aspects of the cell types (e.g. boundaries of reconstructed areas) make estimation impossible.

*2) Show cross-validation on the link prediction*.

Done.

*3) As given, the cell type identity analysis is difficult to parse; the circos-style plots don't tell much and the other plots are often inadequately labelled. It is also unclear as to how validation was performed or was to be observed. Simply seeing ROC curves (or reported areas) or something similar is needed. The figures are of low informational value. How would some well-known supervised method do at this task in cross-validation? It would seem, intuitively, to be a natural choice for learning cell identity*. *See, for example, PMID: 21154911, for a well-defined analysis*.

We have abandoned the Circos-style connectivity plots that, while visually appealing, failed to do an adequate job conveying type and distance-related connectivity. We instead now use a linear plot of connectivity as a function of distance. We feel this more clearly demonstrates the distance-dependent nature of connectivity. We now discuss the possibility and limitations of supervised methods.

*Show ROC style prediction on the cell types. This can be done by cross-validation i.e., do the clustering, then label all the cells using, e.g., 2/3 of the human-labelled data. Then assign each unknown cell a label based on its co-membership with cells of known labels. This would end up giving a ranked score, since probabilistic. The comparison method would be something like nearest-neighbor assignment. To avoid making this a multi-class prediction problem, one could do it one class at a time and not worry about overlapped assignments. This would set a good standard for methods comparison*.

To address this concern we have expanded the set of metrics we use to quantify clustering considering ARI, completeness, and homogeneity. ROC style predictions would be interesting and doable, if only the datasets were larger. As it stands there are only a very small number of examples of many types and removing more for analysis would weaken our power. We feel that this kind of concern is also well addressed by our new and much more comprehensive analysis of the properties of the algorithm.

*4) Consideration and discussion about the edge effects in the retinal data*.

*The retina data set covers a relatively small volume of the retina (on the order of 100 microns in each direction) and most of the cells whose cell bodies are located within this volume have their neurites “cut off”, i.e. they are not fully contained within the volume. This will lead to edge effects that may impact or bias connection patterns, for example cells whose cell bodies are near the center of the field of view should have more synaptic contacts (more partners in the contact matrix) simply because they are more fully captured in the reconstruction. How have such potential edge effects been accounted for in the present analyses? Could there be improvement in the performance of the clustering algorithm by taking into account such biases*?

Done.

*5) Chip example was considered less critical as compared to providing more testing and further details and discussion on the retinal and* C.elegans *examples*.

We somewhat shortened the chip example. We feel it is still useful as it gets the reader to understand that this general class of algorithms could be useful beyond neuroscience.

*6) The use of connectivity information for characterizing nodes in a neural network has a long history in cognitive and systems neuroscience. It would be nice to point this out. A potential starting point is the review of Passingham et al. (2002) in Nature Reviews Neuroscience*.

We added extensive citations to this part of the literature in a newly created Discussion paragraph.

*7) Additional comments regarding unhelpful/unclear figure aspects for algorithm specifics*:

*Are there any known inherent biases in terms of preferences in cluster sizes? It appears, from*
Figure 4*, that there are a small number of fairly large (and quite uniform) clusters and a larger number of quite small ones (that seem quite noisy with respect to the anatomist's cell classification)*.

Our model uses a nonparmetric prior which generally believes that a “small” number of clusters can be used to predict the data, but (should the data support it) but is easily dominated by the likelihood of the data. The good agreement in both cluster size and type with synthetic data leads us to believe that this is not an effect of the prior. Additionally, we perform hyperparameter inference over the cluster concentration parameter alpha. This avoids over-biasing the model towards large clusters. Thus we believe the “noise” present in the clusterings truly reflects uncertainty in the data.

*It is difficult to judge just by visual inspection alone how “good” the agreement between the automated cluster analysis and the anatomist's ordering scheme really is (*Figure 3*, outer ring). Is there a more quantitative way of determining the “fit” of the model with the a priori anatomical cell type assignments*?

As mentioned above, we added additional fit metrics. We also expanded the discussion of this in the text to make it easier to digest for the readers.

*In a similar vein, the text claims (in the subsection headed “Recovering spatial connectivity in multiple graphs simultaneously”) that for* C. elegans *the clusters correspond roughly homogenously to moto, sensory and interneurons, but looking at*
Figure 5*, it is not so clear. Interneurons are found in almost all of the clusters, and very few are truly homogeneous with respect to these three types of neurons. Again, it would be good to think of a way to quantify “homogeneity”*.

We added a quantification of homogeneity and toned down the discussion in the text.

*In the circular plots (e.g.*
Figure 1*), it is not quite clear what the shaded arcs in the center refer to. Is the shading proportional to some sort of density (of what?), and what does the width of these arcs refer to*?

Per above, we have abandoned the circos-style plots, replacing them with (what we hope are) more intuitive and carefully-annotated linear figures.